# The Group-Algebraic Formalism of Quantum Probability and Its Applications in Quantum Statistical Mechanics

**DOI:** 10.3390/e27010059

**Published:** 2025-01-10

**Authors:** Yan Gu, Jiao Wang

**Affiliations:** 1Department of Modern Physics, University of Science and Technology of China, Hefei 230026, China; 2Department of Physics and Fujian Provincial Key Laboratory of Low Dimensional Condensed Matter Physics, Xiamen University, Xiamen 361005, China; phywangj@xmu.edu.cn; 3Lanzhou Center for Theoretical Physics, Lanzhou University, Lanzhou 730000, China

**Keywords:** quantum probability theory, quantum statistical mechanics, group-theoretical characteristic function, quantum-optical master equation, cat map

## Abstract

We show that the theory of quantum statistical mechanics is a special model in the framework of the quantum probability theory developed by mathematicians, by extending the characteristic function in the classical probability theory to the quantum probability theory. As dynamical variables of a quantum system must respect certain commutation relations, we take the group generated by a Lie algebra constructed with these commutation relations as the bridge, so that the classical characteristic function defined in a Euclidean space is transformed to a normalized, non-negative definite function defined in this group. Indeed, on the quantum side, this group-theoretical characteristic function is equivalent to the density matrix; hence, it can be adopted to represent the state of a quantum ensemble. It is also found that this new representation may have significant advantages in applications. As two examples, we show its effectiveness and convenience in solving the quantum-optical master equation for a harmonic oscillator coupled with its thermal environment, and in simulating the quantum cat map, a paradigmatic model for quantum chaos. Other related issues are reviewed and discussed as well.

## 1. Introduction

Probability signifies our ignorance of the exact cause of an event. The theory of probability, as a mathematical branch, emerged in the seventeenth century. According to Laplace [1], probability can be determined as the ratio of favorable outcomes to all equally possible alternatives. But by 1900, this classical interpretation of probability had become outdated and a new statistical interpretation gradually became prevalent. In this new interpretation, it does not make sense to talk of the probability of a single event. Indeed, as exemplified by statistical mechanics, probability is defined in terms of the distribution of an ensemble in the phase space [2], or, in a statistical sense, as frequencies with reference to infinite sequences of repeatable events [3].

The modern mathematical axiomatic framework of probability theory was proposed by Kolmogorov in 1933 [4]. In Kolmogorov’s probability theory, the probability space is defined by a triple (Ω,Σ,μ), where Ω is an arbitrary set (called a sample space), Σ is a Boolean σ-algebra of subsets of Ω (called measurable sets), and μ is a probability measure on Σ [5]. A random variable associated with this probability space is defined as a measurable function {ω∈Ω→f(ω)∈R} in the sample space Ω with the probability distribution function $F\left(x\right)=\int_{f(\omega )\leq x}d\mu \left(\omega \right)$.

When quantum mechanics took shape around 1925–1927, the notion of probability with its statistical interpretation was introduced into quantum mechanics by Born [6], by which the implications of a wave function of a quantum system are successfully decoded. It is well known that the orthodox mathematical formalism of quantum theory proposed by von Neumann [7] is quite different from the Kolmogorovian model. In von Neumann’s theory, states and observables are represented with density matrices and self-adjoint operators in a Hilbert space, respectively. Consequently, the probability space of a quantum system can be defined by a triple $\left(H,P\left(H\right),\hat{\rho }\right)$, where H is a separable Hilbert space, P(H) is the set of all projection operators in the closed subspaces of H, and $\hat{\rho }$ is a density matrix in H. A random variable associated with this probability space is defined as a self-adjoint operator $\hat{X}$ in the Hilbert space H with its spectral representation $\hat{X}=\int \lambda d\hat{E}\left(\lambda \right)$, $\hat{E}\left(\lambda \right)\in P\left(H\right)$, and the expectation value $\langle \hat{X}\rangle =\mathrm{Tr}\left[\hat{\rho }\hat{X}\right]=\int \lambda \mathrm{Tr}\left[\hat{\rho }\hat{E}\left(\lambda \right)\right]d\lambda$ (p. 649 in Ref. [8]).

To understand the transition from classical to quantum theory of the mathematical model used for describing statistical mechanics, as just mentioned, Accardi’s viewpoint [9,10] is advisable: “In analogy with geometry, one should look at probability theory not as the study of the laws of chance but of the possible, mutually inequivalent models for the laws of chance, the choice among which for the description of natural phenomena, being a purely experimental question” (For a physical understanding of quantum-to-classical transition, see Ref. [11]). A crucial point should be emphasized here: the laws of chance for studying indeterministic (statistical) phenomena are as important as the laws of dynamics for studying deterministic (causal) motion of a physical system because they are two different kinds of laws in nature. The former is intrinsically nonlocal, while the latter only describes the locally connected phenomena.

We know that, in classical mechanics, it is well founded that Euclidean spacetime, as the conventional kinetic arena of Newtonian mechanics, has to be substituted with Minkowski spacetime in relativistic mechanics. In geometry, both Newtonian spacetime and Minkowski spacetime are treated as special cases of a Riemanian manifold. In particular, the former has a positive definite metric, while the latter has an indefinite metric. By analogy, it is natural to ask whether it is possible to construct a mathematical model of the probability theory for studying statistical mechanics such that both the Kolmogorovian model and von Neumann’s model can be dealt with in a unified mathematical framework. Indeed, a generalized version of von Neumann’s framework for quantum mechanics has been proposed by Segal [12], who proposed to consider the $C^{*}$-algebra generated by all bounded observables of a quantum system as the primary subject of this theory. In this abstract algebraic formalism, the notion of Hilbert space only plays a secondary role as the representation space of the algebra, and the states of a quantum statistical ensemble (i.e., density matrices in the Hilbert space) are represented by normalized positive linear functionals on the algebra so that the probability space of a quantum system can be represented with (A,ω(a)), where A is a $C^{*}$-algebra and ω(a) is a normalized positive linear functional on A. In fact, using algebraic formalism, the algebraic structure of observables provides a useful framework for analyzing quantum statistical mechanics, particularly for a system in equilibrium state [13], as well as for the study of quantum field theory [14]. Moreover, the fact that one can handle all inequivalent representations simultaneously in the algebraic approach is a manifestation of its power superior to von Neumann’s Hilbert-space approach. In addition, the algebraic approach also allows us to deal with commutative and noncommutative observable algebra within a single mathematical framework.

However, for studying quantum statistical mechanics, the mathematical framework based on the $C^{*}$-algebraic approach has a serious disadvantage, i.e., the disability to involve unbounded observables ubiquitous in quantum mechanics. As such, we have to relax the topological constraints imposed on the observables and use ∗-algebra instead of $C^{*}$-algebra to represent their algebraic structure, as some mathematicians have tried to develop the theory of quantum probability [15]. However, the algebraic approach of quantum probability, though very powerful mathematically, adds little to physical understanding compared with the conventional formulation of quantum theory. Thus, it is desirable to search for natural operational principles that may lead to a specific set of “preferred observables” important for a physical understanding. In this connection, the conventional algebraic (as well as von Neumann’s) approach has ignored a priori the issue of the “preferred observables”.

In this paper, we study a certain class of algebraic probability spaces denoted as (A,ω), with A being a group algebra. We find that when A is generated from a Lie group *G* with a bi-invariant Haar measure—its Lie algebra consists of the set of “preferred observables” of a quantum system—the characteristic function extensively used in the classical probability theory can be extended to the group-algebraic probability space as a normalized non-negative definite function in the group *G*. Furthermore, we find that this group-theoretical characteristic function (GCF) can be adopted to replace the density matrix commonly used in quantum mechanics to represent a state of the quantum ensemble, and more than that, this GCF representation has some significant advantages for applications in quantum statistical mechanics. It is worth noting that, importantly, as shown in [16], where this group-theoretical formalism was first proposed, certain properties of a quantum system are actually rooted in the specific representation of its phase group, i.e., group *G* associated with the group-algebraic probability space on the state vector space of the system. Our study therefore suggests that it is possible to obtain an important physical understanding of a quantum system through its algebraic structure.

The remaining sections of this paper are organized as follows. First of all, the basics of the algebraic formulation of quantum probability will be briefly introduced in Section 2. Then, in Section 3, we will discuss in detail the application of the group-algebraic formalism in statistical mechanics. Next, In Section 4, two examples will be presented to show how the GCF representation may facilitate solving the evolution of a quantum system. Finally, we will summarize our study in Section 5. The mathematical analysis of the group-algebraic formulation of the quantum probability theory, while essential to our discussion, is more technical and will therefore be detailed in Appendix A.

## 2. Algebraic Probability Spaces

Let A be a ∗-algebra, i.e., an algebra over the set of complex numbers C with involution ∗ and unit $1_{A}$. By definition, a linear functional ω(a) of a∈A, taking values in C, is (i) *positive* if $\omega (a^{*}a)\geq 0$, ∀a∈A; (ii) *normalized* if $\omega (1_{A})=1$; and (iii) *a state* if ω is positive and normalized. If ω is a state, we have$$\omega \left(a^{*}\right)=\bar{\omega (a)}\;\;\mathrm{and}\;\;{\left|\omega \left(a^{*}b\right)\right|}^{2}\leq \omega \left(a^{*}a\right)\omega \left(b^{*}b\right),\;\forall a,b\in A.$$
We denote the space of all linear functionals of A by $A^{\prime }$ and the set of all states by S. Thus S is a convex subset of $A^{\prime }$ and its extreme points are called “pure states”.

The pair (A,ω) for ω∈S is called an algebraic probability space, where a real random variable is represented by a real element $a=a^{*}\in A$ and ω(a) is interpreted as the expectation value of the random variable *a* in the state ω.

It should be remarked [15,17] that the identification of ω(a) with the expectation value of a random variable a∈A seems quite formal so far because it is not obvious what the relevant probability distribution (if any) is, which we need to evaluate the expectation value. To show that this is not the case, let us consider a real random variable a∈A and its moment sequence $m_{k}=\omega \left(a^{k}\right)$, where k=0,1,…,N. As for any set $\{c_{k}\in C$; k=0,1,…,N} and $b=\sum_{k=0}^{N}c_{k}a^{k}$ we have$$\omega \left(b^{*}b\right)=\sum_{j,k}^{N}\bar{c_{j}}c_{k}m_{j+k}\geq 0,$$
Then, according to Hambuger’s theorem for the classical moment problem [18], there exists at least one probability measure $\mu_{\omega }^{a}$ such that(1)$$m_{k}=\omega \left(a^{k}\right)=\int_{-\infty }^{\infty }x^{k}\mu_{\omega }^{a}\left(dx\right).$$
Furthermore, if function $\sum_{k}m_{k}z^{k}/k!$ is analytic in the neighborhood of z=0, the probability measure $\mu_{\omega }^{a}$ is uniquely determined.

While this approach using moments is useful for finding a suitable probability measure satisfying Equation (Equation 1) for a single random variable *a*, it is frustrating when we try to generalize it to the case with a set of pairwise commuting random variables for finding their joint probability measure [18]. The conventional way out of this difficulty is to use the Hilbert-space representation of the ∗-algebra A. In this way, first, we need to construct a representation $A∋a\to \hat{\pi }\left(a\right)$ on a Hilbert space H such that a real element *a* is mapped to a selfadjoint operator $\hat{\pi }\left(a\right)$ and each density matrix $\hat{\rho }$ in H is represented by a state $\omega_{\rho }\in S$ satisfying $\omega_{\rho }\left(a\right)=\mathrm{Tr}\left[\hat{\rho }\hat{\pi }\left(a\right)\right]$. Then, we have [19]$$\hat{\pi }\left(a\right)=\int xd{\hat{P}}^{a}\left(x\right),\;\;\;\mu_{\rho }^{a}\left(dx\right)=\mathrm{Tr}\left[\hat{\rho }d{\hat{P}}^{a}\left(x\right)\right],\;\;\mathrm{and}\;\;\omega_{\rho }\left(a\right)=\int_{-\infty }^{\infty }x\mu_{\rho }^{a}\left(dx\right),$$
where $d{\hat{P}}^{a}\left(x\right)$ is the projection valued measure (PVM) of the self-adjoint operator $\hat{\pi }\left(a\right)$.

In fact, when the ∗-algebra A in the probability space (A,ω) is a $C^{*}$-algebra (a normed ∗-algebra with $∥a^{*}{a∥=∥a∥}^{2}$, ∀a∈A), we have the following GNS (after Gelfand, Neumark, and Segal) construction theorem (p. 85 in Ref. [20]):

**Theorem** **1.**
*Let A be a unital $C^{*}$-algebra and ω be a pure state of A. There exists a triple $(H_{\omega },|\psi_{\omega }\rangle ,\pi_{\omega })$, where $H_{\omega }$ is a Hilbert space, $|\psi_{\omega }\rangle \in H_{\omega }$ is a unit vector, and $\pi_{\omega }$: $A\to B(H_{\omega })$ is a ∗-homomorphism from a∈A to a bounded operator ${\hat{\pi }}_{\omega }\left(a\right)$ in $H_{\omega }$, such that (i) $|\psi_{\omega }\rangle$ is a cyclic vector, namely, ${\hat{\pi }}_{\omega }\left(A\right)|\psi_{\omega }\rangle$ is dense in $H_{\omega }$; (ii) $\langle \psi_{\omega }|{\hat{\pi }}_{\omega }\left(a\right)|\psi_{\omega }\rangle =\omega \left(a\right),\;\forall a\in A$; and (iii) the representation $A∋a\to {\hat{\pi }}_{\omega }\left(a\right)$ is irreducible.*


We call this representation of ∗-algebra, A, the cyclic representation associated with the state ω (or the cyclic vector $|\psi_{\omega }\rangle$). Noting that all bounded symmetric operators are self-adjoint and any non-zero vector in $H_{\omega }$ is cyclic with its associated representation unitarily equivalent to $A∋a\to {\hat{\pi }}_{\omega }\left(a\right)$, therefore, when A is a $C^{*}$-algebra, GNS construction provides a Hilbert-space representation $A∋a\to {\hat{\pi }}_{\omega }\left(a\right)$ such that any real element $a=a^{*}\in A$ is mapped to a self-adjoint operator and any density matrix $\hat{\rho }$ in $H_{\omega }$ can be represented by a state $\omega_{\rho }\left(a\right)=\mathrm{Tr}\left[\hat{\rho }{\hat{\pi }}_{\omega }\left(a\right)\right]$.

The GNS construction of a Hilbert-space representation can be generalized to the case where the ∗-algebra is not a $C^{*}$-algebra. Let $D_{\omega }$ be a pre-Hilbert space (a dense subspace of a Hilbert space) and L($D_{\omega }$) be the set of linear operators from $D_{\omega }$ into itself which admits adjoints so that L($D_{\omega }$) becomes a ∗-algebra. Let $|\psi_{\omega }\rangle \in D_{\omega }$ be a unit vector, then there exists a GNS representation $\pi_{\omega }$ of (A,ω) such that (i) $\pi_{\omega }$: $A\to L(D_{\omega })$ is a ∗-homomorphism; (ii) ${\hat{\pi }}_{\omega }\left(A\right)|\psi_{\omega }\rangle =D_{\omega }$; and (iii) $\langle \psi_{\omega }|{\hat{\pi }}_{\omega }\left(a\right)|\psi_{\omega }\rangle =\omega \left(a\right),\;\forall a\in A$. However, in contrast with the $C^{*}$ case, now ${\hat{\pi }}_{\omega }\left(a\right)$ can be an unbounded operator in $D_{\omega }$ and it is difficult to identify the precise conditions for the unbounded symmetric operator ${\hat{\pi }}_{\omega }\left(a\right)$ in $D_{\omega }$ being essentially self-adjoint in the sense of the Hilbert space theory (p. 37 and p. 310 in Ref. [17]).

In the following section, we will propose a special class of algebraic probability spaces, (A,ω), with A being a group algebra. Our analysis will show that, in this mathematical group-algebraic framework, the quantum generalization of the characteristic functions extensively used in the classical probability theory can be realized in a natural way. Moreover, the representation of the ∗-algebra, A, can be generated directly from the representation of the phase group *G* without using the more sophisticated GNS technique. In addition, for a statistical system described with the group-algebraic probability space, its states may have some remarkable features relevant to the group representation.

## 3. Algebraic Probability Spaces as the Mathematical Foundation of
Classical and Quantum Statistical Mechanics

The primary subject for statistical theory of classical mechanics is to address the microscopic motion of molecules in gases. The kinetic theory established to this end appeared in the middle of the nineteenth century, featuring probability that is completely extraneous to the deterministic Newtonian mechanics. It was brought to a mature stage by Clausius, Maxwell, and Boltzmann, with explicit recognition that the random motion of molecules can be characterized with probability distributions of the positions and velocities of individual molecules, and thermal energy is nothing but the kinetic energy of the microscopic motion.

Later, an important contribution to the mathematical framework of classical statistical mechanics was made by Gibbs. He introduced the concept of “statistical ensemble” in the phase space spanned by pairs of canonical variables, in contrast with probability distributions of positions and velocities. Taking Kolmogorov’s measure-theoretical approach to probability theory, Gibbs adopted the phase space as the sample space of random variables. The advantage is that the volume element of the phase space is invariant under the dynamical evolution of the system, and hence it owns the property of an invariant measure.

It was totally unanticipated until the 1930s that probability would also play a crucial role in quantum mechanics deemed to describe the microscopic world. The statistical interpretation of the wave function of a pure quantum state was first introduced by Born [6] and this interpretation was further developed by von Neumann into a mathematically rigorous theory by recognizing that states and observables are the most essential concepts of quantum mechanics. In von Neumann’s theory, states and observables of a quantum system are represented with state vectors (rays) and self-adjoint operators, respectively, in a Hilbert space, while the square of the modulus of the inner product of state vectors provides the probability introduced by Born. Furthermore, von Neumann also pointed out that for establishing the probability theory as the theory of frequencies, it is necessary to introduce the concept of quantum statistical ensemble represented by density matrices in the Hilbert space (p. 298 in Ref. [7]).

Note that although it is still a subject of debate if the quantum probability model proposed by von Neumann can be regarded as a special case of Kolmogorov’s measure-theoretical probability model, now it has been well established, with both physically verified quantum theory and rigorous mathematical criteria, that in nature, one can find some sets of data concerning mutually incompatible sets of observables of quantum systems that can be described with von Neumann’s model rather than Kolmogorov’s model [9].

As mentioned in the Introduction, a remarkable generalization and extension of von Neumann’s mathematical framework is due to Segal [12], who proposed to consider as the primary subject, the $C^{*}$-algebra generated by bounded observables in the mathematical framework of quantum mechanics. However, for studying quantum statistical mechanics, the mathematical framework based on this algebraic approach suffers from the following problems: (i) Unbounded observables are common in quantum mechanics. As such, we have to relax the topological constraints and use ∗-algebra instead of $C^{*}$-algebra to represent the algebraic structure of observables, so that the statistical behavior of a quantum system can be described with the probability space outlined in Section 2; (ii) For all specific quantum theories, they commonly proceed with the input of a classical theory that tells what the relevant observables are and how they are related. In this connection, the algebraic approach (as well as von Neumann’s approach) dispenses, a priori, with the choice of the “preferred observables” altogether; (iii) The Poisson structure of phase space plays a crucial role in classical statistical mechanics, but the corresponding specific structure in the algebraic approach of the mathematical framework of quantum statistical mechanics has not been explicitly explored yet.

To solve problems (ii) and (iii), more restrictive assumptions are needed to impose on the specific formalism of the algebraic probability space. There are various possible routes to this end. One was proposed by one of the authors in Ref. [21], where it was suggested to restrict the ∗-algebra of an algebraic probability space to a specific group algebra $A_{G}$ (denoted by $D^{\prime }\left(G\right)$ for compact Lie groups and $E^{\prime }\left(G\right)$ for locally compact Lie groups, respectively, in Appendix A). The resultant formalism was referred to “the group-theoretical formalism” [16]. Here, we will review, discuss in more detail, and improve this formalism by laying it on a more solid mathematical foundation.

Let us assume that the quantum system of interest is characterized with a basic set of independent dynamical variables $\left\{X_{1},X_{2},\ldots ,X_{n}\right\}$ that forms a basis of a Lie algebra g. We denote by *G* the Lie group generated by g and $A_{G}$ as the corresponding group algebra. Then, the statistical behavior of this quantum system can be described based on the algebraic probability space associated with the group algebra $A_{G}$.

For the applications of this group-theoretical formalism in studying quantum statistical mechanics, we would like to emphasize the following notable aspects.

(a) Determining the “preferred observables” of the system.

Let *X* be an element of the Lie algebra g and $p\left(X\right)\in A_{G}$ be a real polynomial function of *X*. For an arbitrary unitary representation of *G* in a Hilbert space H, the element $p\left(iX\right)\in A_{G}$ will be mapped to a self-adjoint operator in the Hilbert space H. In this case, it is natural to assume that the set of dynamical variables $\left\{iX_{1},iX_{2},\ldots ,iX_{n}\right\}$ play the role of “preferred observables” of the system.

(b) Investigating the evolution of the system.

For the case where the relevant Lie group *G* admits a bi-invariant Haar measure dg, let $G∋g\to \hat{U}\left(g\right)$ be an irreducible unitary representation in a Hilbert space H and $\hat{\rho }$ a density matrix in H. Then, for each $\hat{\rho }$ in H, there exists a group-theoretical characteristic function (GCF) $\varphi_{\rho }\left(g\right)=\mathrm{Tr}\left[\hat{\rho }\hat{U}\left(g\right)\right]$ with the well-defined inverse transform $\hat{\rho }=\int_{G}\varphi_{\rho }\left(g\right)\hat{U^{†}}\left(g\right)dg$ (see Appendix A). In other words, after giving an irreducible unitary representation of *G* in H, the state of the quantum system, conventionally represented with the density matrix $\hat{\rho }$ in H, can be represented with the corresponding GCF, $\varphi_{\rho }$. Since $\varphi_{\rho }\left(g\right)$ is a c-number (infinitely differentiable function on *G*), it is advisable to represent the dynamical evolution equation of the system as differential equations of $\varphi_{\rho }\left(g\right)$ instead, as it is generally easier to manipulate than the corresponding equation in terms of the density matrix, especially when the dimension of the representation space H is large. In addition, by using the GCF representation, we can handle all inequivalent representations of the group algebra $A_{G}$ simultaneously in the same mathematical approach.

(c) Dealing with composite quantum systems.

Let $S_{1}$ and $S_{2}$ be two quantum systems described with probability spaces associated with group algebras, $A_{G_{1}}$ and $A_{G_{2}}$, respectively. Consider the composite system $S=S_{1}+S_{2}$ described with the probability space $\left(A_{G},\varphi \left(g\right)\right)$, where $G=G_{1}⨂G_{2}$ and φ(g) is a state on $A_{G}$. The reduced state φ(g) for subsystems $S_{1}$ and $S_{2}$ is now simply given by $\varphi_{1}\left(g_{1}\right)=\varphi \left(g_{1}⨂e_{2}\right)$ and $\varphi_{2}\left(g_{2}\right)=\varphi \left(e_{1}⨂g_{2}\right)$, respectively, where $g_{j}\in G_{j}$ and $e_{j}$ is the identity of $G_{j}$ for j=1,2. Furthermore, in group-theoretical formalism, the state of the composite system φ(g) is separable with respect to subsystems $S_{1}$ and $S_{2}$ if and only if $\varphi \left(g\right)=\sum_{k=1}^{n}p_{k}\kappa_{k}\left(g_{1}\right)\omega_{k}\left(g_{2}\right)$, where $p_{k}>0$, $\sum_{k=1}^{n}p_{k}=1$, and $\left\{\kappa_{k}\left(g_{1}\right),\omega_{k}\left(g_{2}\right),k=1,\ldots ,n\right\}$ are states on $A_{G_{1}}$ and $A_{G_{2}}$, respectively [22].

(d) Revealing the quantum-to-classical transition.

Let g be the Lie algebra of a connected Lie group *G*. We take its dual vector space $g^{*}$ as the phase space of a classical mechanical system. Let $E(g^{*})$ be the space of all $C^{\infty }$ functions on $g^{*}$. It is known that $E(g^{*})$ admits a Poisson structure defined by linear Poisson brackets [23]:(2)$${\left\{f,k\right\}}_{\xi }=\sum_{\alpha ,\beta }\left(\xi ,\left[X_{\alpha },X_{\beta }\right]\right)\frac{\partial f}{\partial \xi_{\alpha }}\frac{\partial k}{\partial \xi_{\beta }}=\sum_{\alpha ,\beta ,\gamma }C_{\alpha ,\beta }^{\gamma }\xi_{\gamma }\frac{\partial f}{\partial \xi_{\alpha }}\frac{\partial k}{\partial \xi_{\beta }},\;\;\;\;\;f,k\in E\left(g^{*}\right),$$
where $\xi \in g^{*},\;\left\{X_{\alpha },\alpha =1,\ldots ,n\right\}$ is a basis of g, and $\xi_{\alpha }=\left(\xi ,X_{\alpha }\right)$ and $C_{\alpha ,\beta }^{\gamma }$ are the structure constants of g. The Lie–Poisson phase space $g^{*}$ can be foliated into a family of leaves, i.e., coadjoint orbits of *G* (p. 226 in Ref. [24]), where each leaf is a *G*-invariant symplectic manifold (p. 20 in Ref. [25]). These symplectic leaves are called superselection sectors of the phase space $g^{*}$.

To discuss the applications of the group-algebraic formalism, in the following, we will focus on the quantum systems of canonical variables. First, we show that the statistical behavior of these systems can be described using the algebraic probability spaces associated with the Heisenberg–Weyl group [26]. To this end, let us consider a quantum system with canonical variables (q,p) whose commutation relation is $[\hat{q},\hat{p}]=i\hbar$. This commutation relation can be regarded as the first of the following three pairs of Lie brackets of a three-dimensional Lie algebra g with basis $\left\{X_{1},X_{2},X_{3}\right\}$$$\left[X_{1},X_{2}\right]=X_{3},\;\;\left[X_{1},X_{3}\right]=0,\;\;\mathrm{and}\;\;\left[X_{2},X_{3}\right]=0,$$
by identifying $X_{1}$, $X_{2}$, and $X_{3}$ with $\hat{q}$, $\hat{p}$, and iℏ, respectively. Exponentiating the Lie algebra g, we obtain the Heisenberg–Weyl group H(1), of which each element g∈H(1) can be written as exp(a·X) with $a\in R^{3}$. Moreover, the multiplication law of group elements is given by exp(a·X)exp(b·X)=exp(c·X), with(3)$$c=(a_{1}+b_{1},a_{2}+b_{2},a_{3}+b_{3}-\frac{1}{2}\left(a_{1}b_{2}-a_{2}b_{1}\right)).$$

Now, let us turn to the classical systems associated with the Heisenberg–Weyl group H(1). Let $g^{*}$ be the dual space of g, $\xi \in g^{*}$, and $\xi_{j}=\left(\xi ,X_{j}\right)$. Noting that the coadjoint action of g=exp(a·X)∈H(1) on the vector $\xi \in g^{*}$ is infinitesimally generated by$$Ad_{a\cdot X}^{*}\xi =\left(\xi ,\left[X,a\cdot X\right]\right)=\left(a_{2}\xi_{3},-a_{1}\xi_{3},\;0\right),$$
we can conclude that there are two classes of coadjoint orbits in $g^{*}$: (i) the two-dimensional planes perpendicular to axis $\xi_{3}$ when $\xi_{3}\neq 0$ and (ii) all points on the plane of $\xi_{3}=0$. According to Equation (Equation 2), the Poisson brackets on the two-dimensional coadjoint orbits can be written as$${\left\{f,k\right\}}_{\xi }=\left(\frac{\partial f}{\partial \xi_{1}}\frac{\partial k}{\partial \xi_{2}}-\frac{\partial f}{\partial \xi_{2}}\frac{\partial k}{\partial \xi_{1}}\right)\xi_{3},\;\;\;\;\;f,k\in E\left(g^{*}\right).$$
Substituting $\xi_{1}$ and $\xi_{2}$ with *q* and *p*, respectively, and considering the special symplectic leaf $\xi_{3}=1$, we immediately obtain the standard expressions of Poisson brackets, the crucial structure of classical mechanics.

(e) Quantization of classical systems of compact phase spaces by constructing the corresponding algebraic probability space.

The compactness of the phase space implies that, for the corresponding quantized system, the number of phase cells in the phase space, or equivalently, the dimension of the state vector space H, is a finite integer. As an illustrating example, let us consider a system whose degree of freedom is one and the phase space is a two-dimensional torus. Without loss of generality, we suppose that the torus has a unit area with a unit period in both dimensions. As such, the dimension of H is N=1/(2πℏ). Following the scheme suggested in Ref. [27], the first step of quantization is to construct a pair of conjugate orthonormal bases in the vector space H, {|j〉} and $\left\{|\bar{l}\rangle \right\}$, with j,l=0,…,N−1, such that$$|\bar{l}\rangle =\frac{1}{\sqrt{N}}\sum_{j=0}^{N-1}e^{i2\pi jl/N}|j\rangle ,\;\;\;|j\rangle =\frac{1}{\sqrt{N}}\sum_{l=0}^{N-1}e^{-i2\pi jl/N}|\bar{l}\rangle .$$
Next, we can define a pair of canonical variables $\left(\hat{q},\hat{p}\right)$ in H, by assuming that {|j〉} and $\left\{|\bar{l}\rangle \right\}$ are their eigenstates, respectively, i.e., $\hat{q}|j\rangle =\frac{j}{N}|j\rangle ,\;j=0,\ldots ,N-1$ and $\hat{p}|\bar{l}\rangle =\frac{l}{N}|\bar{l}\rangle ,\;l=0,\ldots ,N-1$. It is worth noting that the the operator pair $\left(\hat{q},\hat{p}\right)$ thus defined does not obey the conventional Heisenberg commutation relation for the flat two-dimensional phase space; rather, it follows the Weyl commutation rule:$$e^{i2\pi n\hat{p}}e^{i2\pi n\hat{q}}=e^{i2\pi nm/N}e^{i2\pi n\hat{q}}e^{i2\pi n\hat{p}},\;\;\;\left(m,n\right)\in Z^{2}.$$

Now, let us consider a compact Lie group *G*, any element of which g∈G is represented with three parameters, {m,n,z}, where $\left(m,n\right)\in Z^{2}$, 0≤m,n<N, and 0≤z<2π. The multiplication law is$$\left\{m_{1},n_{1},z_{1}\right\}\left\{m_{2},n_{2},z_{2}\right\}=\left\{m_{1}+n_{1},m_{2}+n_{2},z_{1}+z_{2}+\frac{\pi }{N}\left(m_{2}n_{1}-m_{1}n_{2}\right)\right\}.$$
Comparing this multiplication law with that of the Heisenberg–Weyl group H(1) (see Equation (Equation 3)), the group *G* can be regarded as a discretized version of H(1). Noting that the Weyl commutation relation gives a projective representation of *G* in the state vector space H, we can therefore lift this projective representation to an irreducible unitary representation $G∋g\to \hat{U}\left(g\right)=e^{i(z+\pi nm/N)}e^{i2\pi m\hat{q}}e^{i2\pi n\hat{p}}$. Thus, using the quantization procedure mentioned above, the behavior of the quantized system can be described with the algebraic probability space associated with the group algebra $A_{G}$.

## 4. Application Examples

As discussed in the previous section, the group-theoretical formalism provides not only a more self-consistent platform conceptually to address the problems of quantum statistical mechanics, but also one more optional tool for solving the problems technically. Indeed, in some cases, it may greatly facilitate calculations. In the following part of this section, we will illustrate its superiority with two examples. One is a harmonic oscillator coupled with a thermal environment; its relaxation process to the equilibrium state will be discussed. Another is the cat map, a well-known paradigm for chaos. We will discuss its relaxation process characterized by the coarse-grained entropy.

The group-theoretical formalism can be advantageous in other situations as well. For example, it has been found to be much more convenient than the conventional methods for solving the problem of quantum Brownian motion. See Ref. [28] for a detailed study based on the Caldeira–Leggett master equation and its extension to the Lindblad type.

### 4.1. The Harmonic Oscillator in a Thermal Environment

The evolution of a quantum system that interacts with its environment is of fundamental importance in the theory of open quantum systems [29,30]. A harmonic oscillator moving irreversibly in its thermostatic surroundings is presumably the simplest model for probing this issue. The reduced density matrix ρ of the oscillator as a function of time is assumed to follow the quantum-optical master equation(4)$$\frac{d\rho }{dt}=-i\omega \left[a^{†}a,\rho \right]+\Gamma \left(N_{\beta }+1\right)\left(2a\rho a^{†}-\left\{a^{†}a,\rho \right\}\right)+\Gamma N_{\beta }\left(2a^{†}\rho a-\left\{aa^{†},\rho \right\}\right).$$
Here, Γ is the damping constant, $\beta =1/(k_{B}T)$ with $k_{B}$ and *T* being the Boltzmann constant and the environmental temperature, respectively, $N_{\beta }=1/\left(e^{\hbar \beta \omega }-1\right)$ is the expected number of thermal quanta, and$$a=\frac{1}{\sqrt{2\hbar M\omega }}\left(M\omega q+ip\right),\;\;\;\;a^{†}=\frac{1}{\sqrt{2\hbar M\omega }}\left(M\omega q-ip\right).$$

This master equation was first derived by Louisell [31] for addressing a damped mode of the radiation field in a cavity and was treated by Haake [32] as an example of the generalized master equation constructed by Nakajima and Zwanzig. It plays a central role not only in the theoretical interpretations of many quantum-optical experiments [33,34], but also in revealing the environment-induced decoherence of open quantum systems [29].

In the following, we will show how to solve Equation (Equation 4) with the GCF representation and scrutinize the relaxation process of a pure quantum state towards a mixed state. For the pair of canonical variables (q,p), as discussed in Section 3 (d), their commutation relation [q,p]=iℏ can be seen as a pair of Lie brackets of a three-dimensional Heisenberg-Weyl Lie algebra. Denote the nilpotent Lie group generated by this Lie algebra as H(1) and let$$g\to U\left(g\right)=e^{i(xp+yq+z\hbar )},\;g\in H\left(1\right)$$
be an irreducible unitary representation of H(1) on the state vector space H of the system, where $\{\left(x,y\right)\in R^{2}$, 0≤z<2π/ℏ} represents a canonical system of coordinates on the group H(1), the GCF of the density matrix ρ(t) of the system at time *t* is (see Section 3 (b) and Appendix A)$$\varphi \left(t,g\right)={\mathrm{Tr}}_{H}\left[\rho \left(t\right)U\left(g\right)\right]=e^{iz\hbar +v(x,y,t)},$$
where v(x,y,t) is a complex-number function. Consequently, the inverse mapping has the form$$\rho \left(t\right)=\int_{H(1)}\varphi \left(t,g\right)U^{†}\left(g\right)dg,$$
where $dg={\left(\hbar /2\pi \right)}^{2}dxdydz$ is the bi-invariant Haar measure on H(1). If we use complex variables ($a,a^{†}$) instead of (q,p), the unitary representation of H(1) in H has the form $U\left(g\right)=e^{ua^{†}-\bar{u}a+iz\hbar }$ with $u=u_{1}+iu_{2}=-\sqrt{\frac{\hbar M\omega }{2}}x+i\sqrt{\frac{\hbar }{2M\omega }}y$, which gives an alternative expression for the GCF$$\varphi \left(t,g\right)={\mathrm{Tr}}_{H}\left[\rho \left(t\right)U\left(g\right)\right]=e^{iz\hbar +v(u,t)}.$$

Using formulae $aU\left(g\right)=\left(-\frac{\partial }{\partial \bar{u}}+\frac{u}{2}\right)U\left(g\right)$ and $a^{†}U\left(g\right)=\left(\frac{\partial }{\partial u}+\frac{\bar{u}}{2}\right)U\left(g\right)$, the quantum-optical master Equation (Equation 4) can be converted into the following equation for v(u,t):(5)$$\frac{\partial v(u,t)}{\partial t}=-\{\left[\left(\Gamma u_{1}+\omega u_{2}\right)\frac{\partial }{\partial u_{1}}+\left(\Gamma u_{2}-\omega u_{1}\right)\frac{\partial }{\partial u_{2}}\right]v\left(u\right)+\left(1+2N_{\beta }\right){\Gamma |u|}^{2}\},$$
or, alternatively, for v(x,y,t)(6)$$\frac{\partial v(x,y,t)}{\partial t}=\left[\left(\frac{y}{M}-\Gamma x\right)\frac{\partial }{\partial x}-\left(M\omega^{2}x+\Gamma y\right)\frac{\partial }{\partial y}\right]v\left(x,y\right)-\frac{\hbar }{2}\left(1+2N_{\beta }\right)\Gamma \left(M\omega x^{2}+\frac{y^{2}}{M\omega }\right).$$
In the following, we discuss the solution to these two equations with two different initial conditions, respectively.

(a) Case 1: An initial Gaussian state

When the initial state of the system is a Gaussian state centered at $\left(\bar{q},\bar{p}\right)$, i.e.,$$\langle q|\psi_{G}\rangle ={\left(\frac{1}{2\pi \sigma^{2}}\right)}^{\frac{1}{4}}e^{-\frac{{\left(q-\bar{q}\right)}^{2}}{4\sigma^{2}}+\frac{i}{\hbar }\bar{p}\left(q-\bar{q}\right)},$$
the GCF of the initial density matrix ρ(0) reads $\varphi \left(0,g\right)=\langle \psi_{G}\left|U\left(g\right)\right|\psi_{G}\rangle =e^{iz\hbar +v(x,y,0)}$, where$$v\left(x,y,0\right)=i\left(x\bar{p}+y\bar{q}\right)-\left(\frac{\hbar^{2}x^{2}}{8\sigma^{2}}+\frac{\sigma^{2}y^{2}}{2}\right).$$
Then, it is straightforward to write down the solution of the master Equation (Equation 6) as$$v\left(x,y,t\right)=A\left(t\right)x+B\left(t\right)y+a\left(t\right)x^{2}+b\left(t\right)y^{2}+c\left(t\right)xy$$
with$$\left\{\begin{array}{c} A\left(t\right)=ie^{-\Gamma t}\left(\bar{p}\cos\omega t-M\bar{q}\omega \sin\omega t\right),\\ B\left(t\right)=ie^{-\Gamma t}\left(\bar{q}\cos\omega t+\frac{\bar{p}}{M\omega }\sin\omega t\right),\\ a\left(t\right)=-\frac{\hbar M\omega }{4}\left(2N_{\beta }+1\right)\left(1-e^{-2\Gamma t}\right)-\frac{1}{2}e^{-2\Gamma t}\left(\frac{\hbar^{2}}{4\sigma^{2}}\cos^{2}\omega t+M^{2}\sigma^{2}\omega^{2}\sin^{2}\omega t\right),\\ b\left(t\right)=-\frac{\hbar }{4M\omega }\left(2N_{\beta }+1\right)\left(1-e^{-2\Gamma t}\right)-\frac{1}{2}e^{-2\Gamma t}\left(\sigma^{2}\cos^{2}\omega t+\frac{\hbar^{2}}{4M^{2}\sigma^{2}\omega^{2}}\sin^{2}\omega t\right),\\ c\left(t\right)=\frac{e^{-2\Gamma t}}{8M\sigma^{2}\omega }\left(4M^{2}\sigma^{4}\omega^{2}-\hbar^{2}\right)\sin2\omega t. \end{array}\right.$$

In the GCF representation, the purity of the system has the form [16]P(t)=Tr[ρ(t)2]=∫H(1)|φ(t,g)|2dg=ℏ2π∫dx∫dye2Re[v(x,y,t)],
and thus the linear entropy [35] of the system is(7)$$\begin{array}{c} S\left(t\right)=-\ln P\left(t\right)=\ln\frac{2}{\hbar }\sqrt{4a\left(t\right)b\left(t\right)-c{\left(t\right)}^{2}}\;\;\;\;\;\;\;\;\;\;\;\;\;\;\;\;\;\;\;\;\;\;\;\;\;\;\;\;\;\;\;\;\;\;\;\;\;\;\;\;\;\;\;\;\;\;\;\;\;\;\\ \;\;\;\;\;=\frac{1}{2}\ln\left[e^{-4\Gamma t}+{\left(1-e^{-2\Gamma t}\right)}^{2}{\left(2N_{\beta }+1\right)}^{2}+2\left(1-e^{-2\Gamma t}\right)e^{-2\Gamma t}\left(2N_{\beta }+1\right)\cosh2r\right], \end{array}$$
where the squeeze factor *r* is defined by $e^{-r}=\frac{\sigma }{\sigma_{c}}$ with $\sigma_{c}=\sqrt{\frac{\hbar }{2M\omega }}$. As shown elsewhere [36], for a classical harmonic oscillator with an initial localized phase-space distribution, the evolution of its entropy depends on neither its environmental temperature nor its initial position in the phase space. Therefore, it is somehow perplexing to comprehend such differences between both quantum and classical dynamical behavior in the context of non-equilibrium statistical mechanics.

From Equation (Equation 7), we can see that if the initial state is a coherent state with r=0,(8)$$S\left(t\right)=\ln[1+2N_{\beta }\left(1-e^{-2\Gamma t}\right)],$$
suggesting that the entropy increases monotonically from S(0)=0 to its equilibrium value $S_{eq}=\ln\left(1+2N_{\beta }\right)$, as expected. However, for an initial squeezed state with r>0, the evolution of entropy becomes more complicated. There exists a certain time$$t_{m}=\frac{1}{2\Gamma }\ln\left[\frac{2+4N_{\beta }+4N_{\beta }^{2}-2\left(1+2N_{\beta }\right)\cosh2r}{\left(1+2N_{\beta }\right)\left(1+2N_{\beta }-\cosh2r\right)}\right]$$
before which, S(t) increases in time but after which S(t) decreases and converges to $S_{eq}$ (see Figure 1). Since $t_{m}$ increases from $\frac{\ln2}{2\Gamma }$ to *∞* as $N_{\beta }$ increases from 0 to $N_{c}=\sinh^{2}r$, such an interesting non-monotonic “hump” characteristic of the time curve of S(t) appears if and only if $N_{\beta }<N_{c}$.

(b) Case 2: An initial Fock state

When the initial state is a Fock state, i.e., an energy eigenstate of the isolated harmonic oscillator, denoted as |n〉, it is convenient to take the complex coordinate system on H(1) instead and consequently, the GCF can be written as$$\varphi \left(0,g\right)=\langle n|U\left(g\right)|n\rangle =e^{iz\hbar +\frac{{\left|u\right|}^{2}}{2}}\langle n|e^{-\bar{u}a}e^{ua^{†}}|n\rangle =e^{iz\hbar +v_{n}\left(u\right)}.$$
Noting that for the coherent state |α〉, we have $\langle n|\alpha \rangle =\frac{\alpha^{n}}{\sqrt{n!}}e^{-\frac{{\left|\alpha \right|}^{2}}{2}}$ and as a result,$$\langle n|e^{-\bar{u}a}e^{ua^{†}}|n\rangle =\frac{1}{\pi n!}\int d^{2}\alpha {\left|\alpha \right|}^{2n}e^{-{\left|\alpha \right|}^{2}+u\bar{\alpha }-\bar{u}\alpha }\;\;\;\;\;\;\;\;\;\;\;\;\;\;\;$$=e−|u|2πn!∑m=0nnm∫dα1α12me−(α1−iu2)2∫dα2α22(n−m)e−(α2+iu1)2,
where $\alpha =\alpha_{1}+i\alpha_{2}$. Using the formula $\int_{-\infty }^{\infty }dxx^{n}e^{-{\left(x-y\right)}^{2}}=\frac{\sqrt{\pi }}{{\left(2i\right)}^{n}}H_{n}\left(iy\right)$, we have(9)vn(u)=v(u,0)=ln[(−1)nn!22ne−12|u|2∑m=0nnmH2m(u2)H2(n−m)(u1)],
where $H_{n}\left(x\right)$ is the Hermite polynomial of degree *n*.

The solution of the master Equation (Equation 5) with the initial condition (Equation 9) is therefore(10)$$v\left(u,t\right)=v_{n}\left(e^{-\Gamma t}u\right)+\frac{1}{2}\left(1+2N_{\beta }\right)\left(e^{-2\Gamma t}-1\right){\left|u\right|}^{2}$$
and the linear entropy of the system readsS(t)=−ln[1π∫du1∫du2e2Re[v(u,t)]].
Making use of Equation (Equation 10) and letting $a^{2}=1+\left(1+2N_{\beta }\right)\left(e^{2\Gamma t}-1\right)$, the entropy of the system (with initial state ρ(0)=|n〉〈n|) can be written explicitly as(11)Sn(t)=−ln[e2Γt(n!22n)2π∑m1,m2nnm1nm2IP(m1,m2)IP(n−m1,n−m2)],
where [37](12)IP(m1,m2)=∫dxe−a2xH2m1(x)H2m2(x)=22(m1+m2)Γ(m1+m2+12)(1−a2)m1+m2a2(m1+m2)+1F12(−2m1,−2m2,12−m1−m2,a22(a2−1)).
From Equations (Equation 11) and (Equation 12), we know that$$S_{n}\left(0\right)=0,\;S_{n}{\left(t\right)}_{t\to \infty }=S_{eq}=\ln\left(1+2N_{\beta }\right).$$
We also note that the expression of $S_{0}\left(t\right)$ given by Equation (Equation 11) is the same as that given by Equation (Equation 8), as the ground state of a harmonic oscillator itself is a coherent state.

On the other hand, the expressions of $S_{n}\left(t\right)$ for n>0 become more and more complicated as *n* grows. Taking the case of n=1 as an example, we have$$S_{1}\left(t\right)=\ln\left[\frac{{\left[e^{-2\Gamma t}+\left(1-e^{-2\Gamma t}\right)\left(1+2N_{\beta }\right)\right]}^{3}}{e^{-4\Gamma t}+{\left(1-e^{-2\Gamma t}\right)}^{2}{\left(1+2N_{\beta }\right)}^{2}}\right].$$

To appreciate the role played by the parameter $N_{\beta }$, we consider the extreme points of curve $S_{1}\left(t\right)$. They are the roots of $dS_{1}\left(t\right)/dt=0$, which are given by$$t_{\pm }=\frac{1}{2\Gamma }\ln\left[\frac{1+2N_{\beta }-2N_{\beta }^{2}-4N_{\beta }^{3}\pm \sqrt{1+6N_{\beta }+10N_{\beta }^{2}-8N_{\beta }^{4}}}{\left(N_{\beta }-1\right){\left(1+2N_{\beta }\right)}^{2}}\right].$$
So a necessary condition for $S_{1}\left(t\right)$ having extreme points is $N_{\beta }<1+d_{1}$ with $d_{1}=\frac{1}{2}\left(\sqrt{3}-1\right)$∼0.366. Thus, $S_{1}\left(t\right)$ is non-decreasing only when $N_{\beta }$ is greater than the critical value $N_{c}\left(1\right)=1+d_{1}$. For $1<N_{\beta }<N_{c}\left(1\right)$, $S_{1}\left(t\right)$ has two extreme points located at $t=t_{-}$ and $t=t_{+}$, respectively, satisfying $S\left(t_{-}\right)>S_{eq}>S\left(t_{+}\right)$. As $N_{\beta }$ decreases from $N_{c}\left(1\right)$ to 1, $t_{+}$ increases to infinity, so that $S_{1}\left(t\right)$ exhibits only one hump when $N_{\beta }<1$. In Figure 2, $S_{1}\left(t\right)$, for a set of different values of $N_{\beta }$, is shown for comparison. Note that in contrast with the case where the initial state is Gaussian, here, we have an additional phase featuring two extremes or two turning points. In this phase, S(t) first increases and then decreases at $t=t_{-}$ until $t=t_{+}$ when it turns up and eventually saturates.

Finally, an interesting observation is that, although the analytical expressions of $S_{n}\left(t\right)$ for larger *n* are more complicated, qualitatively, $S_{n}\left(t\right)$ is still characterized by the three-phase feature that is the same as $S_{1}\left(t\right)$, i.e., a “hump” phase, a phase of two turning points, and a monotonically increasing phase.

To summarize, for the two initial conditions investigated, thanks to the GCF, we can solve the motion equations explicitly, based on which the relaxation process of the system can be analyzed in detail. See Ref. [36] for more and detailed discussions.

### 4.2. The Cat Map

The cat map describes the evolution of a periodically kicked particle with a compact phase space. The Hamiltonian of the system is [38]$$H=\frac{1}{2}p^{2}+\frac{K}{2}q^{2}\delta_{1}\left(t\right),$$
where, without loss of generality, both the particle mass and the kicking period have been assumed to be unity. The kicks are exerted instantaneously and accordingly; $\delta_{1}\left(t\right)$ represents a sequence of δ functions spaced uniformly in time. In addition, the periodic boundary conditions of a unit period are imposed on both *q* and *p*, making the phase space a unit torus. Integrating the equations of motion over a unit time from just before the *k*th kick to just before the (k+1)th kick, we have the cat map$$\left(\begin{array}{c} q_{k+1}\\ p_{k+1} \end{array}\right)=\left(\begin{array}{cc} 1-K & 1\\ -K & 1 \end{array}\right)\left(\begin{array}{c} q_{k}\\ p_{k} \end{array}\right)\;\left(\mathrm{mod}\;1\right).$$
Note that *K* is the only parameter of the map. If |K−2|>2, the map is an Anosov diffeomorphism on the torus and the motion it generates is strongly chaotic (in particular, the motion is mixing and ergodic). The Arnold cat map discussed in the following corresponds to K=−1 [39].

Based on the quantization scheme outlined in Section 3 (e) and the quantum counterpart of the classical Hamiltonian given above, the unitary time evolution operator corresponding to the classical cat map can be integrated out exactly, which is$$\hat{M}=e^{-i\pi N{\hat{p}}^{2}}e^{-i\pi NK{\hat{q}}^{2}},$$
with the second and the first factor on the r.h.s. being responsible for the kick and the free motion between two successive kicks, respectively. In terms of density operator, the evolution of the quantum system can be expressed as(13)$${\hat{\rho }}^{k+1}=\hat{M}{\hat{\rho }}^{k}{\hat{M}}^{†}.$$
Obviously, based on this quantum map directly, it is difficult for both analytical and numerical study of quantum evolution, especially when the dimension of the Hilbert space N=1/(2πℏ) is large.

However, if we take advantage of the group-theoretical formalism, the quantum map can be greatly simplified [40,41,42]. Setting $\hat{U}\left(m,n\right)=e^{i\pi mn/N}e^{i2\pi m\hat{q}}e^{i2\pi n\hat{p}}$, $\left(m,n\right)\in Z^{2}$, the GCF for a given density operator $\hat{\rho }$ is defined as$$\varphi \left(m,n\right)=\mathrm{Tr}[\hat{\rho }\hat{U}\left(m,n\right)]$$
and the inverse transformation is$$\hat{\rho }=\frac{1}{N}\sum_{m=m_{0}}^{m_{0}+N-1}\sum_{n=n_{0}}^{n_{0}+N-1}\varphi \left(m,n\right){\hat{U}}^{†}\left(m,n\right).$$
It suggests that a quantum ensemble can be assigned equivalently by the GCF φ(m,n) over any N×N square lattice (note that $m_{0}$ and $n_{0}$ are two arbitrarily chosen integers).

By substituting the GCF into Equation (Equation 13), the corresponding quantum evolution can be rewritten as$$\varphi^{k+1}\left(m_{k+1},n_{k+1}\right)=\varphi^{k}\left(m_{k},n_{k}\right)$$
with$$\left(\begin{array}{c} m_{k+1}\\ n_{k+1} \end{array}\right)=\left(\begin{array}{cc} 1 & K\\ -1 & 1-K \end{array}\right)\left(\begin{array}{c} m_{k}\\ n_{k} \end{array}\right),$$
implying that φ(m,n) is simply permuted on a finite N×N lattice. As a consequence, even for a large *N*, the quantum evolution problem can be dealt with conveniently.

For the cat map, it is straightforward to show that both its purity and the linear entropy do not vary in time. In order to appreciate how its classical mixing and ergodic dynamics affects the quantum motion, we therefore turn to its coarse-grained entropy instead:$$S_{\epsilon }^{k}=-\ln\left(\mathrm{Tr}\left[{\left({\hat{\rho }}_{\epsilon }^{k}\right)}^{2}\right]\right)=-\ln\left(\sum_{m=m_{0}}^{m_{0}+N-1}\sum_{n=n_{0}}^{n_{0}+N-1}|\varphi_{\epsilon }^{k}\left(m,n\right){|}^{2}\right).$$
If a Gaussian coarse-graining factor with width ϵ is taken, then for Nϵ≫1, we have [41]$$\varphi_{\epsilon }^{k}\left(m,n\right)\approx e^{-\pi^{2}\epsilon^{2}\left(m^{2}+n^{2}\right)}\varphi^{k}\left(m,n\right).$$
From this expression, we can tell that the role played by the coarse-graining factor is a wave filter such that $\varphi_{\epsilon }^{k}\left(m,n\right)$ can be negligible when (m,n) is far from the origin (0,0).

For an initial Gaussian wavepacket of width *a* that is centered at $\left(q_{c},p_{c}\right)$, we have$$\varphi^{0}\left(m,n\right)=e^{i2\pi (mq_{c}+np_{c})}e^{-a^{2}\pi^{2}\left(m^{2}+n^{2}\right)}.$$
The system evolves in terms of the GCF, and its coarse-grained entropy as a function of time is shown in Figure 3. We can see that the coarse-grained entropy increases linearly at the initial stage, signaling the underlying mixing dynamics. After this relaxation stage, the coarse-grained entropy stops increasing but shows some fluctuation patterns due to the quantum coherent effect. As N→∞, the quantum coherent effect tends to decay and the quantum coarse-grained entropy converges to its classical counterpart as expected. For more detailed discussions and interesting related research on the Wigner distribution and the quantum scar and anti-scar effects, see Ref. [42].

Finally, we would like to mention that, for the numerical computations presented in Figure 3, the cost is the same for any *N* value. It does not increase as *N* increases, and this is a remarkable advantage of the GCF representation in this case. In contrast, if the simulations were carried out based on the density matrix, the cost would increase as ∼$N^{3}$ instead, which would be prohibitively expensive with the computing resources available today.

## 5. Summary

We have briefly reviewed the mathematical framework of quantum statistical mechanics and the quantum probability theory developed by mathematicians in recent years, in an attempt to address the connection between them. It is worth noting that the latter is a broad field encompassing directions developed with various mathematical assumptions. Here, we have mainly focused on the theories based on $C^{*}$- and ∗-algebraic representations of the probability space, particularly analyzing their pros and cons from the perspective of physics. We therefore propose to restrict the ∗-algebra to a specific group algebra where the commutation relations of the dynamical variables of the system are integrated. In particular, we have established the quantum characteristic function in a more mathematically rigorous way than was previously studied in Ref. [16]. The properties of this group-algebraic representation have been discussed in detail, suggesting that it is not only more self-consistent conceptually but also more convenient in certain applications.

To illustrate how to use this suggested group-theoretic formalism in analytical and numerical studies and to appreciate its effectiveness, two application examples have been presented. Obviously, searching for more applicable situations is desirable and should be interesting for future studies. 

## Figures and Tables

**Figure 1 entropy-27-00059-f001:**
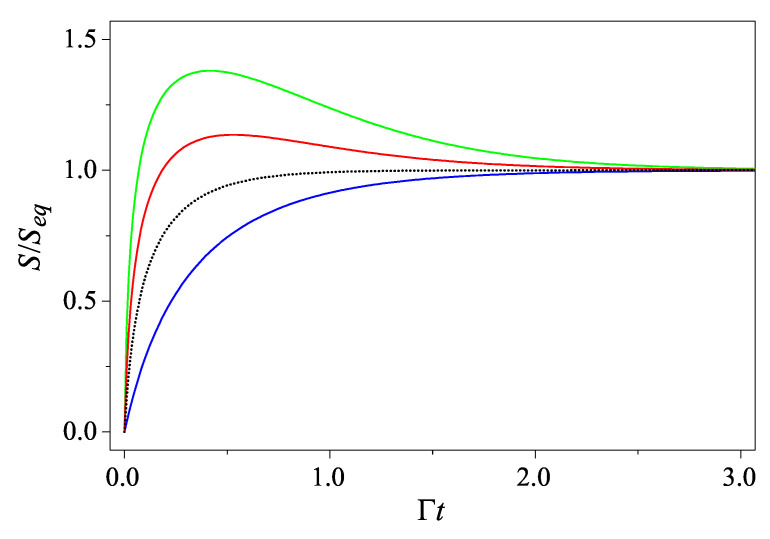
The entropy as a function of time for an initial Gaussian state with $N_{\beta }=1$. The four curves, from top to bottom, are for $N_{c}=5.5$ (green), 2.5 (red), 1 (black dotted), and 0 (blue), respectively. The black dotted line is for the critical case that $N_{\beta }=N_{c}$ and the blue line is for the coherent state with r=0.

**Figure 2 entropy-27-00059-f002:**
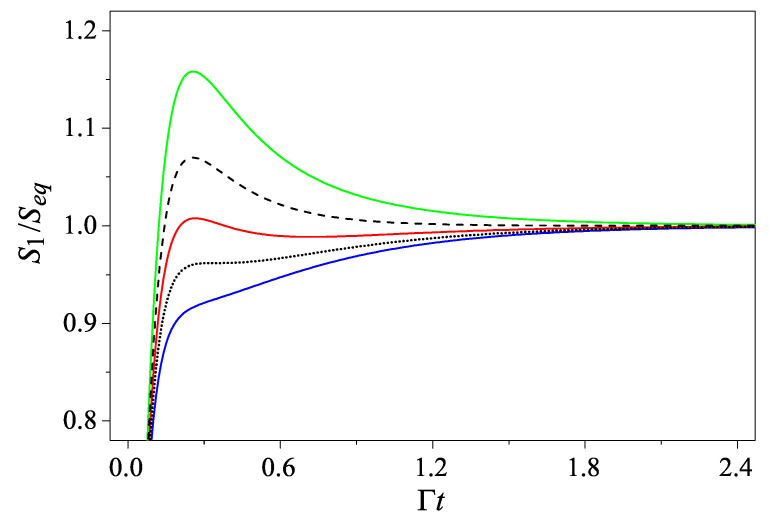
The entropy $S_{1}\left(t\right)$ for the initial state |1〉 with $N_{c}\left(1\right)=\left(\sqrt{3}-1\right)/2\approx 1.366$. The five curves, from top to bottom, are for $N_{\beta }=0.82$ (green), 1.0 (black dashed), 1.18 (red), $N_{c}\left(1\right)$ (black dotted), and 1.6 (blue), respectively. Note that the three phases $S_{1}\left(t\right)$ may undergo are separated by the two critical curves, the black dashed and the black dotted line, respectively.

**Figure 3 entropy-27-00059-f003:**
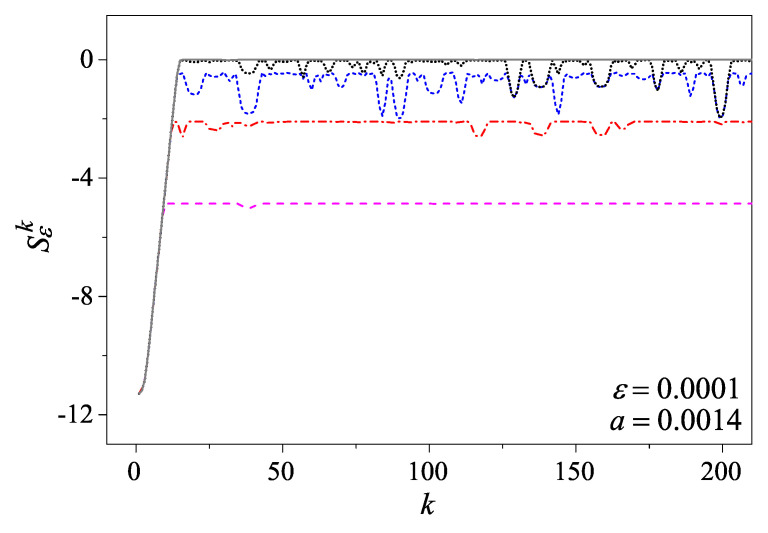
The coarse-grained entropy for the quantum cat map. The initial Gaussian distribution is centered at (0, 0) with a=0.0014. The Gaussian coarse-graining parameter is ϵ=0.00004. The curves, from bottom to top, are for $N=10^{5}$ (dashed, magenta), $4\times 10^{5}$ (dot-dashed, red), $10^{6}$ (short-dashed, blue), and $2\times 10^{6}$ (dotted, black), respectively. As a comparison, the gray solid line is for the corresponding coarse-grained entropy for the classical cat map of the same settings.

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
