# Peer review of "The Group-Algebraic Formalism of Quantum Probability and Its Applications in Quantum Statistical Mechanics"

_entropy, 2025, doi:10.3390/e27010059_

Round 1

Reviewer 1 Report

Comments and Suggestions for Authors

The authors present a method for solving the problems of quantum statistical mechanics. This method involves using a group theory approach and restricting the relevant observables. The group theoretical formalism provides not only a more self-consistent way to address the problems of quantum statistical mechanics but also gives a new tool for solving technical problems. The paper gives two examples. One is a harmonic oscillator coupled with a thermal environment. Another is a cat map which is a well-known paradigm for chaos. They also discuss methods for solving the problem of quantum Brownian motion. This is a well-written article and I recommend publication.

Comments on the Quality of English Language

The use English language is excellent.

Author Response

Referee's comment: The authors present a method for solving the problems of quantum statistical mechanics. This method involves using a group theory approach and restricting the relevant observables. The group theoretical formalism provides not only a more self-consistent way to address the problems of quantum statistical mechanics but also gives a new tool for solving technical problems. The paper gives two examples. One is a harmonic oscillator coupled with a thermal environment. Another is a cat map which is a well-known paradigm for chaos. They also discuss methods for solving the problem of quantum Brownian motion. This is a well-written article and I recommend publication.

Our reply: We are grateful for Referee's positive comment. Thanks!

Reviewer 2 Report

Comments and Suggestions for Authors

The authors try to provide the interplay between three fields of mathematics/physics/probability, that is Statistical mechanics, quantum probability and representation theory. The author wrote:

"We show that the theory of quantum statistical mechanics is a special model in the framework of the quantum probability theory developed by mathematicians, by extending the characteristic function in the classical probability theory to the quantum probability theory. As dynamical variables of a quantum system must respect certain commutation relations, we take the group generated by a Lie algebra constructed with these commutation relations as the bridge, so that the classical characteristic function defined on a Euclidean space is transformed to a normalized, nonnegative definite function defined on this group. Indeed, on the quantum side, this group-theoretical characteristic function is equivalent to the density matrix, hence can be adopted to represent the state of a quantum ensemble. It is also found that this new representation may have significant advantages in applications."

The framework of "algebraic probability space", which seems to be the starting point of the authors, concerns with a pair made of a, often unital, involutive algebra A and a positive functional f (i.e. f(x*x)geq0 for all x in A). In all intersting cases, A is a mere involutive algebra without any reasonable topology (see the case of the CCR algebra for which density of positions and momenta are considered). In such a situation, the Gelfand Naimark Segal representation associated to the positive functional f provides, in general, unbounded operators. Unfortunately, although this is a completely expected fact, most of the experts in Quantum Probability tend to bypass this very crucial (and slippery) point. A list (far to be complete) of papers concerning this point is:

-Araki, Shiraishi Publ. Res. Inst. Math. Sci. 7 (1971/72), 105–120.

-Araki  Publ. Res. Inst. Math. Sci. 7 (1971/72), 121-152.

-Crismale, Duvenhage, Fidaleo Anal. Math. Phys. 11 (2021), no. 1, Paper No. 11

-Powers Comm. Math. Phys. 21 (1971), 85–124

-Powers Trans. Amer. Math. Soc. 187 (1974), 261–293

-Gudder  Pacific J. Math. 80 (1979), 141-149.

Gudder, Scruggs Pacific J. Math. 70 (1977), 369-382.

The reviewer is in favor of the publication, but only after a quite deep discussion concerning the point relative to "algebraic probability space" "states" and "GNS representations" as explained above.

Author Response

Referee’s comment

The authors try to provide the interplay between three fields of mathematics/physics/probability, that is Statistical mechanics, quantum probability and representation theory. The author wrote:

 "We show that the theory of quantum statistical mechanics is a special model in the framework of the quantum probability theory developed by mathematicians, by extending the characteristic function in the classical probability theory to the quantum probability theory. As dynamical variables of a quantum system must respect certain commutation relations, we take the group generated by a Lie algebra constructed with these commutation relations as the bridge, so that the classical characteristic function defined on a Euclidean space is transformed to a normalized, nonnegative definite function defined on this group. Indeed, on the quantum side, this group-theoretical characteristic function is equivalent to the density matrix, hence can be adopted to represent the state of a quantum ensemble. It is also found that this new representation may have significant advantages in applications."

The framework of "algebraic probability space", which seems to be the starting point of the authors, concerns with a pair made of a, often unital, involutive algebra A and a positive functional f (i.e. f(x*x)geq0 for all x in A). In all intersting cases, A is a mere involutive algebra without any reasonable topology (see the case of the CCR algebra for which density of positions and momenta are considered). In such a situation, the Gelfand Naimark Segal representation associated to the positive functional f provides, in general, unbounded operators. Unfortunately, although this is a completely expected fact, most of the experts in Quantum Probability tend to bypass this very crucial (and slippery) point. A list (far to be complete) of papers concerning this point is:

-Araki, Shiraishi Publ. Res. Inst. Math. Sci. 7 (1971/72), 105–120.

-Araki  Publ. Res. Inst. Math. Sci. 7 (1971/72), 121-152.

-Crismale, Duvenhage, Fidaleo Anal. Math. Phys. 11 (2021), no. 1, Paper No. 11

-Powers Comm. Math. Phys. 21 (1971), 85–124

-Powers Trans. Amer. Math. Soc. 187 (1974), 261–293

-Gudder  Pacific J. Math. 80 (1979), 141-149.
-Gudder, Scruggs Pacific J. Math. 70 (1977), 369-382.

The reviewer is in favor of the publication, but only after a quite deep discussion concerning the point relative to "algebraic probability space" "states" and "GNS representations" as explained above.

Our reply to Referee’s comment

In this manuscript, our motivation is to discuss a special algebraic model of quantum probability where the *-algebra A of the algebraic probability space is a group algebra AG with G a unimodular locally compact Lie group. The algebraic probability space of this model can be represented as (E'(G), f ), where E'(G) consists of all distributions with compact supports on G and f is a normalized nonnegative definite infinitely differentiable function on G (termed the GCF,see Appendix). In this model: (i) E'(G) is a countably normed topological space and (ii) every representation of G generates a representation of the *-algebra E'(G) (see Ref. [23], p.154).  In other words, we do not use GNS construction to obtain the representation of the *-algebra E'(G).

In the revised manuscript, we have made two modifications to make this point clearer (see lines 152-155 and lines 447-448 highlighted with blue color).

As a consequence, the applicability of this special quantum probability model for studying quantum statistical mechanics is limited. In particular, it cannot be used to treat problems of quantum fields.

We are grateful to Referee for pointing to us that the GNS representation associated to a positive functional f provides, in general, unbounded operators. We think this point and the literatures Referee referred us to are helpful for our further studies to extend this model to a more comprehensive mathematical framework.

Round 2

Reviewer 2 Report

Comments and Suggestions for Authors

Even if the referee is tempted to reject the paper (the authors declared not to know the problem relative to the possible unboundedness of operators appearing in the GNS representation, and thus elegantly skipped about this crucial question), the paper appears well-written and contains results which can be useful for applications. For this motivations, he suggests the publication.